# The Relationships Between Healthcare Access, Gender, and Psychedelics and Their Effects on Distress

**DOI:** 10.3390/healthcare13101158

**Published:** 2025-05-16

**Authors:** Sean Matthew Viña

**Affiliations:** Department of Sociology, The University of the Incarnate Word, 4301 Broadway, San Antonio, TX 78209, USA; vina@uiwtx.edu

**Keywords:** medicine, gender, psychedelics, inequality, distress

## Abstract

**Background**: Structural inequalities in healthcare access may influence how individuals experience the psychological effects of psychedelic substances, potentially limiting positive outcomes among vulnerable populations. **Objectives**: This study uses data from the National Survey on Drug Use and Health (2008–2019; N = 484,732) to examine how public and private health insurance moderate the association between psychedelic use and psychological distress. **Methods:** Ordinary least squares (OLS) regression models indicate that private health insurance is associated with lower psychological distress, while public insurance is associated with higher distress. **Results:** Psychedelic use moderates these associations, reinforcing the protective pattern linked to private insurance and intensifying distress among those with public coverage. These patterns vary by gender: among men, psychedelic use does not significantly alter the association between insurance type and distress; among women, however, psilocybin and lysergic acid diethylamide (LSD) use are associated with lower distress among those with private insurance, but with higher distress among those with public insurance. **Conclusions:** These findings indicate that while psychedelics may interact with existing healthcare conditions, they do not mitigate structural inequalities and may, in some cases, exacerbate them.

## 1. Introduction

Although ample research finds that psychedelic use is associated with better health outcomes [1,2,3,4,5,6,7,8], growing evidence suggests that marginalized populations experience fewer benefits, with disparities observed in relation to race, gender, LGBTQ status, marital status, employment status, and socioeconomic standing [9,10,11,12,13,14,15,16,17]. Yet, one critical factor has been largely overlooked in this body of research: healthcare access. The reason why healthcare likely affects the relationship between psychedelic use and health outcomes is best explained through the theory of Minorities′ Diminished Psychedelic Returns (MDPR). While some propose that biological mechanisms—such as differences in metabolism rates—may contribute to disparities in outcomes associated with psychedelic use [18,19], MDPR offers a sociocultural perspective, asserting that structural inequalities can hinder the positive effects of psychedelics on health before, during, and after use [12]. More specifically, MDPR theory highlights that the effects of psychedelics on health depend not only on an individual′s internal set (mindset, expectations, and mental health history), but also on their external setting, including their social environment, cultural attitudes, and access to supportive resources [20,21]. When individuals face chronic stress, discrimination, or social exclusion, their ability to sustain positive psychedelic outcomes is diminished [22]. In fact, evidence suggests that individuals with smaller social networks or those who are socially isolated—including those that experience isolation due to geographic location (e.g., urban vs. rural environments)—may experience worse health outcomes associated with psychedelic use [22,23]. Overall, MDPR theory suggests that marginalized individuals are faced with less optimal conditions and have access to fewer resources for favorable psychedelic experiences and subsequent health improvements.

While prior MDPR research has explored how identity-based factors, like race, gender, and class, affect psychedelic outcomes, few studies have investigated whether structural aspects of healthcare systems—such as public versus private insurance—play a moderating role. This study builds on and extends MDPR theory by analyzing how healthcare access, as a structural determinant, shapes the association between psychedelic use and psychological distress. Ample evidence finds that access to medical care profoundly influences physical and mental health outcomes. Public healthcare programs such as Medicare, Medicaid, and TRICARE are often linked to poorer health outcomes due to systemic inefficiencies, including long wait times, limited specialist access, and weaker doctor–patient relationships [24,25,26,27]. Medicaid patients, for example, frequently experience shorter consultations and less personalized care, leading to lower satisfaction and decreased adherence to treatment [26,27,28]. Medicare enrollees face high out-of-pocket costs and restricted access to specialists, while TRICARE users contend with network limitations and stigma surrounding mental healthcare, particularly among military personnel [29,30,31,32,33]. Given these challenges, individuals relying on public healthcare already struggle with basic medical needs, making it likely that these barriers extend to their ability to benefit from psychedelic-assisted therapies.

In contrast, private healthcare generally provides better outcomes, due to improved efficiency, faster access, and stronger doctor–patient relationships. Patients with private insurance are more likely to receive timely care, reducing stress and uncertainty [34]. Greater resource availability, including advanced medical technologies and personalized mental health support, further contributes to better long-term health outcomes [35,36,37,38]. Additionally, private healthcare fosters higher patient satisfaction, as individuals report feeling more valued and receiving clearer explanations of their treatment plans [36,39]. However, these advantages are not equally distributed across all demographics, particularly when considering gender disparities in healthcare access and quality.

Gender plays a critical role in shaping healthcare experiences, often compounding disparities in both public and private systems. While men generally report higher satisfaction with private healthcare, due to stronger doctor–patient relationships and more direct treatment [40,41,42], women frequently encounter systemic biases that lead to misdiagnosis, symptom dismissal, and delays in care [43,44,45,46,47,48]. Medical training has historically prioritized male physiology, resulting in poorer diagnostic accuracy for conditions like heart attacks and strokes, which present differently in women [49]. Additionally, women′s symptoms are more likely to be misattributed to psychological conditions, leading to inappropriate mental health diagnoses [50,51]. These disparities persist across both public and private healthcare, though women in public systems experience even greater barriers. While men may already receive optimal benefits from private healthcare, psychedelic use could amplify the advantages of private healthcare for women by helping them to better navigate systemic biases.

Given these inequities, it follows that disparities in healthcare access shape the relationship between psychedelics and health. Individuals in high-quality healthcare settings may be better positioned to integrate psychedelic experiences into a supportive medical environment, maximizing their therapeutic potential. Conversely, those in underfunded public healthcare systems—already struggling with inadequate treatment—may experience fewer benefits, or even turn to psychedelics as a form of self-medication, rather than receiving a structured health intervention. Given that psychedelics have been linked to increased psychological flexibility and self-regulation [52,53], individuals with access to better healthcare may be more capable of translating these effects into sustained health improvements. Meanwhile, those who face structural barriers in healthcare may see limited gains, reinforcing the existing health disparities outlined by MDPR.

The objective of this study is to assess whether healthcare access moderates the association between psychedelic use and psychological distress, and whether these associations differ by gender. Grounded in the Minorities′ Diminished Psychedelic Returns (MDPR) framework, the analysis hypothesizes that individuals with private insurance will exhibit more favorable associations between psychedelic use and psychological distress than those with public insurance, due to systemic differences in healthcare quality and support. It further hypothesizes that these moderation effects will be more pronounced among women, given persistent gender-based disparities in healthcare access, diagnosis, and treatment. To test these hypotheses, the study examines the relationship between psychedelics, psychological distress, healthcare access, and gender using data from the National Survey on Drug Use and Health. The dataset includes 484, 732 participants aged 18 years or older, and spans annual survey waves from 2008 to 2019. The analysis builds on prior methodology for exploring health outcomes related to psychedelic usage based on the NSDUH [6,16,54,55,56]. A series of nested ordinary least squares regression models were conducted in Stata 18 to evaluate the associations between various psychedelics (i.e., MDMA, psilocybin, DMT, ayahuasca, peyote/mescaline, and LSD), the type of healthcare access (public or private), and gender on past-30-day psychological distress.

## 2. Data and Methods

This study used publicly available data from the National Survey on Drug Use and Health (NSDUH), pooled from twelve consecutive annual cross-sectional waves collected between 2008 and 2019 (N = 484,732). The NSDUH is a nationally representative survey administered each year by the Substance Abuse and Mental Health Services Administration (SAMHSA) to measure substance use, mental health outcomes, and healthcare access among the U.S. civilian, non-institutionalized population. To ensure valid national estimates without inflating the population size, the person-level survey weights were divided by the number of years pooled. Table 1 provides descriptive statistics for all the dependent, independent, and control variables used in the analysis. The full NSDUH public-use data files are freely and directly accessible via the SAMHSA data archive: https://www.samhsa.gov/data/data-we-collect/nsduh-national-survey-drug-use-and-health/datafiles/2002-2019 (accessed on 11 January 2024). The NSDUH was initially conducted with approval from the internal review board of the Substance Abuse and Mental Health Services Administration. The present study was reviewed by the author′s institutional ethics committee and deemed exempt from further review, as it relies on de-identified, publicly available data and poses minimal risk to participants. No additional ethics approval or consent was required.

### 2.1. Study Replications

This study follows the methodological approach of previous research examining the relationship between psychedelic use and health outcomes, utilizing data from the National Survey on Drug Use and Health (NSDUH) [2,6,11,12,55,56,57,58,59,60,61,62]. In addition, it incorporates the same variables used in studies that explore the link between psychedelics and mental health [2,6,59], based nearly a dozen other publications that apply established statistical methods to analyze psychedelic use and related outcomes using NSDUH data.

### 2.2. Dependent Variable

Respondents completed the Kessler Psychological Distress Scale (K6) [63,64] to assess their level of distress over the previous month. This instrument measures six feelings or experiences, such as feeling nervous, hopeless, restless, deeply depressed, perceiving everything as an effort, and feeling worthless. The respondents rated each item on a 5-point Likert scale. The scores for these measures were then summed to create a variable representing psychological distress in the past month, ranging from 0 to 24, with higher scores indicating higher levels of distress. The Kessler scale is a widely used and highly reliable measure for assessing psychological distress in individuals with panic disorder, generalized anxiety disorder, bipolar disorder, and schizophrenia [65,66].

### 2.3. Independent Variables

First, gender is a binary variable representing women and men. Next, respondents were asked to indicate whether they had health insurance. Those who had health insurance could then select from five variables to indicate their type of coverage: private health insurance, Medicare, Medicaid, Tricare, and other health insurance. While private care remained a binary variable, Medicare, Medicaid, and Tricare were combined into a single variable called public health insurance (yes vs. no). As discussed earlier, these public programs have been consistently associated with poorer health outcomes, due to systemic inefficiencies such as limited access, care delays, and stigmatization of mental health needs [24,25,26,27,28,29,30,31,32,33]. A sensitivity analysis confirmed that each of these public programs—when examined individually—was independently associated with worse psychological distress outcomes compared to private insurance, justifying their grouping. The category “other insurance” includes respondents who reported Indian Health Service (IHS) coverage, military/veterans′ coverage not included under TRICARE, or other miscellaneous forms of coverage. It was included as a separate control in the regression due to its heterogeneity and low frequency, which made it analytically distinct from both public and private insurance groups.

To assess psychedelic use, respondents were asked whether they had ever used any of the following substances, even once: MDMA, DMT, ayahuasca, psilocybin, LSD, mescaline, and peyote. To maintain consistency with prior research, the six “classic” psychedelics—DMT, ayahuasca, psilocybin, LSD, mescaline, and peyote—were combined into a single measure of lifetime classic psychedelic use [58,59]. Furthermore, recognizing that psychedelic use may vary across sociodemographic and cultural contexts, as reflected in NSDUH data [67,68,69,70,71,72], this study adhered to previous recommendations to analyze each substance separately [73]. Additionally, peyote and mescaline were grouped together due to their common origins [72,74].

### 2.4. Socioeconomic, Demographic, and Drug Use Control Variables

This study incorporated key sociodemographic controls across all regression models. Socioeconomic status was measured using two variables: household income, categorized into seven brackets—less than $10,000, $10,000–$19,999, $20,000–$29,999, $30,000–$39,999, $40,000–$49,999, $50,000–$74,999, and $75,000 or more; and educational attainment, divided into less than a high school diploma, high school graduate, some college, and a college degree or higher. Age was initially structured as a categorical variable, with 18 as the reference category and subsequent groups including 19, 20, 21, 22–23, 24–25, 26–29, 30–34, 35–49, 50–64, and 65 and older. Although age, income, and education are technically categorical, this study followed the recommendations of Long and Freese [75] by testing whether treating them as continuous variables would substantively alter the results. A sensitivity analysis confirmed that whether these variables were treated as categorical or continuous had no meaningful impact on the findings. Therefore, for ease of interpretation, they were treated as continuous in the regression models.

Marital status was categorized into married, widowed, divorced/separated, and single (never married, reference category). Racial/ethnic identity included Black, Hispanic, Asian, Native American or Alaska Native, Native Hawaiian or Pacific Islander, and multiracial individuals, with white respondents serving as the reference group. The number of children in the household was measured as a continuous variable, ranging from 0 to 3 or more. All models included survey year controls, aligning with prior studies analyzing NSDUH data. Given the established relationships between substance use and mental health, the analysis also controlled for lifetime use of cocaine, marijuana, phencyclidine (PCP), inhalants, other stimulants, sedatives, pain relievers, and tobacco products (e.g., smokeless tobacco, pipe tobacco, cigars, and daily cigarette consumption) [2,6,11,12,54,55,56,57,58,59,60,61,62]. The age at first alcohol use was a continuous variable. Thrill-seeking was a combination of two variables that asked whether a respondent enjoyed carrying out “dangerous” and “risky” activities (Cronbach′s α = 0.85). Finally, religiosity was measured through two separate variables. Religious attendance captured the frequency of religious service participation in the past year, categorized as follows: (0) never, (1) 1–2 times, (2) 3–5 times, (3) 6–24 times, (4) 25–52 times, and (5) more than 52 times. Religious salience was an index based on three items: (1) the importance of religious beliefs in one′s life; (2) the extent to which religious beliefs influence personal decisions; and (3) the significance of having religious friends (Cronbach′s α = 0.83).

### 2.5. Analytic Strategy

To confirm data validity and national representativeness, the first step calculated means (for continuous variables) and proportions (for categorical variables). While mean and proportional difference testing verified sample alignment with national estimates, it was not the focus of this study, and should not be overemphasized. This step ensured data robustness before proceeding to the main regression analyses.

Next, weighted means were calculated for continuous variables and proportions for categorical variables, using NSDUH survey weights to ensure national representativeness. A post-estimation LINCOM command was then used in Stata 18 to determine statistical differences between means or proportions across subpopulations [76,77]. While traditional *t*-tests are common for such comparisons, LINCOM allows for linear hypothesis testing based on model-adjusted estimates, offering a more flexible approach in complex survey designs. These differences were computed for all key outcome variables, comparing women to men, individuals with private insurance to those without, and those without public insurance to those with public insurance (Appendix A).

Following this, a series of ordinary least squares (OLS) regression models were used to examine the relationships between gender, insurance status, psychedelic use, and psychological distress. To optimize space, all regression models are presented in Appendix A, with key coefficients visualized in figures within this paper. Appendix A provides the base models: the first model assesses the relationship between MDMA, lifetime classic psychedelic use (LCPU), and psychological distress; the second model introduces health insurance as a factor; the third model examines the main effects of all six psychedelic variables on psychological distress; and the fourth model incorporates health insurance variables. These models are estimated for the full sample (Models 1–4), as well as separately for men (Models 5–8) and women (Models 9–12).

Appendix A presents two-way interaction models assessing the interaction between psychedelic use and private health insurance (Models 1–7), and between psychedelic use and public health insurance (Models 8–14), for the total population. These interactions were included to test whether the psychological effects of psychedelics differed depending on healthcare access—an approach grounded in the MDPR framework, which emphasizes that structural conditions, such as access to medical care, shape the extent to which individuals benefit from psychedelic experiences. To analyze gender differences, an intersectional approach was used, recognizing distinct experiences across groups by running separate models for men and women [76]. Appendix A display the interaction effects (Psychedelic Use × Insurance) separately for men and women. Statistical differences between regression coefficients across gender models were assessed using post-estimation seemingly unrelated regression (SUEST) commands in Stata, which enabled comparisons of estimates between overlapping datasets [78].

To ensure representativeness, the NSDUH weights were adjusted using a scalar factor to align single-year weights with national population estimates. This involved dividing the person-level weights by the number of years pooled (12) to prevent inflation of the national estimates across the combined dataset. All analyses accounted for the complex survey design and incorporated the sampling weights provided in Stata 18. As the NSDUH is a cross-sectional survey, all findings should be interpreted as associations, rather than causal effects.

Unlike previous NSDUH-based psychedelic research that has not accounted for multiple comparisons [9,56,59], this study conducted a sensitivity analysis by applying the Benjamini–Hochberg False Discovery Rate (BH-FDR) adjustment to control for false positives. Given the exploratory nature of the analysis and the large number of statistical tests, BH-FDR helps to balance sensitivity and error control, reducing false discoveries without being overly conservative like the Bonferroni correction [79,80]. This approach follows best practices for large-scale social science and epidemiological research, where detecting real effects while minimizing Type I errors is essential [81,82]. The FDR threshold was set at 0.05, aligning with established guidelines [82]. The sensitivity analysis confirmed that applying the BH-FDR correction did not alter the results, with all significant associations maintaining q-values below 0.05. In addition, this study followed the STROBE (Strengthening the Reporting of Observational Studies in Epidemiology) guidelines for transparent reporting of cross-sectional analyses. Because the K6 scale ranges from 0 to 24, and scores of 13 or higher typically indicate serious psychological distress, even a one-point change is considered clinically meaningful—particularly in nationally representative samples where the average scores are well below this threshold.

## 3. Results

Appendix A show that women report significantly higher psychological distress than men (−0.36, *p* < 0.001; 95% CI: −0.45–−0.26) and are more likely to be uninsured (0.03, *p* < 0.001; 95% CI: 0.03–0.04) and enrolled in public health insurance (−0.06, *p* < 0.001; 95% CI: −0.06–−0.05), while being less likely to have private insurance (0.01, *p* < 0.01; 95% CI: 0.00–0.01). They also report lower lifetime use of all psychedelics (*p* < 0.001), except for ayahuasca, where there is no statistical difference. Compared to men, women have lower family income (−0.30, *p* < 0.001; 95% CI: −0.32–−0.28), slightly higher educational attainment (−0.05, *p* < 0.001; 95% CI: −0.06–−0.04), and greater religiosity (−0.57, *p* < 0.001; 95% CI: −0.59–−0.55). They are also less likely to be single (−0.06, *p* < 0.001; 95% CI: −0.06–−0.05) or married (−0.04, *p* < 0.001; 95% CI: −0.04–−0.04), and more likely to be widowed (−0.06, *p* < 0.001; 95% CI: −0.06–−0.06) or divorced (−0.04, *p* < 0.001; 95% CI: −0.04–−0.03). Women report lower overall drug use, but are more likely to use sedatives (−0.01, *p* < 0.001; 95% CI: −0.01–−0.01) and tranquilizers (−0.03, *p* < 0.001; 95% CI: −0.03–−0.03).

Individuals with private insurance report lower distress (−0.32, *p* < 0.001; 95% CI: −0.41–−0.24) and are significantly less likely to be uninsured (−0.11, *p* < 0.001; 95% CI: −0.12–−0.11) or on public insurance (−0.35, *p* < 0.001; 95% CI: −0.35–−0.34). Those with private insurance are also less likely to have used MDMA (−0.02, *p* < 0.001; 95% CI: −0.02–−0.02), LCPU (−0.03, *p* < 0.001; 95% CI: −0.04–−0.03), psilocybin (−0.03, *p* < 0.001; 95% CI: −0.03–−0.03), DMT (−0.00, *p* < 0.001; 95% CI: −0.00–−0.00), LSD (−0.04, *p* < 0.001; 95% CI: −0.04–−0.04), and mescaline/peyote (−0.02, *p* < 0.001; 95% CI: −0.02–−0.01), but show no difference in ayahuasca use. They tend to have higher family income (1.52, *p* < 0.001; 95% CI: 1.48–1.56), more education (0.34, *p* < 0.001; 95% CI: 0.33–0.35), and are more likely to be married (0.06, *p* < 0.001; 95% CI: 0.06–0.07), while less likely to be single (−0.06, *p* < 0.001; 95% CI: −0.07–−0.06). Their rates of overall drug use are lower across most substances.

Individuals on public insurance have higher distress (0.31, *p* < 0.001; 95% CI: 0.24–0.39) and are more likely to be uninsured (0.35, *p* < 0.001; 95% CI: 0.34–0.35), while being less likely to have private insurance (−0.35, *p* < 0.001; 95% CI: −0.35–−0.34). They report lower use of MDMA (−0.01, *p* < 0.001; 95% CI: −0.01–−0.01), LCPU (−0.02, *p* < 0.001; 95% CI: −0.02–−0.02), psilocybin (−0.02, *p* < 0.001; 95% CI: −0.02–−0.02), LSD (−0.03, *p* < 0.001; 95% CI: −0.03–−0.02), and mescaline/peyote (−0.01, *p* < 0.001; 95% CI: −0.01–−0.01), with no difference in ayahuasca or DMT use. Public insurance holders have lower family income (−1.04, *p* < 0.001; 95% CI: −1.07–−1.00), lower educational attainment (−0.32, *p* < 0.001; 95% CI: −0.33–−0.31), and higher religiosity (−0.26, *p* < 0.001; 95% CI: −0.28–−0.25). They are also more likely to be divorced (0.05, *p* < 0.001; 95% CI: 0.04–0.05) or widowed (0.09, *p* < 0.001; 95% CI: 0.09–0.10), and less likely to be married (−0.07, *p* < 0.001; 95% CI: −0.07–−0.06) or single (−0.07, *p* < 0.001; 95% CI: −0.07–−0.06). Results from these mean and proportional differences align with previous research on distress, gender, and socioeconomic status.

### 3.1. Mainline and Two-Way Interaction Results Among the Total Population

The mainline results (Appendix A) indicate that health insurance status is associated with distress across the total population and by gender. Among the total population (Model 1), private health insurance is associated with lower distress (b = −0.956, *p* < 0.001), while public health insurance is associated with higher distress (b = 0.170, *p* < 0.01). For psychedelic use, among the total population (Model 3), LCPU (b = −0.314, *p* < 0.05) is associated with lower distress, while MDMA (b = 0.207, *p* < 0.05) is linked to higher distress. Additionally, psilocybin (b = −0.361, *p* < 0.05) and peyote/mescaline (b = −0.497, *p* < 0.01) are associated with lower distress, whereas DMT and LSD show no significant associations (Model 4).

While the results indicate that private health insurance, compared to not having private insurance, is associated with lower distress (Figure 1), interaction terms (Appendix A) show that this negative association is strengthened for those who have used lifetime classic psychedelics (b= −0.378, *p* < 0.01 b = −0.378, *p* < 0.01 b = −0.378, *p* < 0.01, Model 1), psilocybin (b= −0.318, *p* < 0.05 b = −0.318, *p* < 0.05 b = −0.318, *p* < 0.05, Model 3), and LSD (b= −0.327, *p* < 0.05 b = −0.327, *p* < 0.05 b = −0.327, *p* < 0.05, Model 7). Conversely, although public health insurance, compared to not having public insurance, is associated with greater distress, interaction terms indicate that this positive association is further amplified among individuals who have used lifetime classic psychedelics (b = 0.936, *p* < 0.001 b = 0.936, *p* < 0.001 b = 0.936, *p* < 0.001, Model 8), MDMA (b = 0.582, *p* < 0.001 b = 0.582, *p* < 0.001 b = 0.582, *p* < 0.001, Model 9), psilocybin (b = 0.923, *p* < 0.001 b = 0.923, *p* < 0.001 b = 0.923, *p* < 0.001, Model 10), peyote/mescaline (b = 0.719, *p* < 0.05 b = 0.719, *p* < 0.05 b = 0.719, *p* < 0.05, Model 13), and LSD (b = 0.796, *p* < 0.001 b = 0.796, *p* < 0.001 b = 0.796, *p* < 0.001, Model 14).

### 3.2. Mainline and Two-Way Interaction Results Among Men and Women

The mainline results are shown in Appendix A. For men (Model 5), private health insurance is linked to lower distress (b = −1.077, *p* < 0.001), and public health insurance is associated with higher distress (b = 0.190, *p* < 0.01). Among women (Model 9), private health insurance is associated with lower distress (b = −0.917, *p* < 0.001), but public health insurance shows no significant association with distress.

The mainline results (Appendix A) show that among men (Model 7), LCPU (b = −0.390, *p* < 0.05) is linked to lower distress, while MDMA (b = 0.193, *p* < 0.10) is marginally associated with higher distress. Among individual psychedelics (Model 8), only peyote/mescaline (b = −0.512, *p* < 0.05) is associated with lower distress, while psilocybin, DMT, and LSD remain non-significant. Among women (Model 11), LCPU (b = −0.303, *p* < 0.05) is linked to lower distress, while MDMA is not significant. Among individual psychedelics (Model 12), psilocybin (b = −0.396, *p* < 0.05) and peyote/mescaline (b = −0.469, *p* < 0.05) are associated with lower distress, whereas DMT and LSD remain non-significant.

Among men (Appendix A), while private health insurance is associated with lower distress and public health insurance with higher distress, the interaction terms do not indicate that psychedelic use moderates these associations. On the other hand, among women (Appendix A), there are more notable interactions between psychedelics and health insurance (Figure 2). While private health insurance is associated with lower distress, this relationship is significantly moderated by psilocybin use (*b* = −0.609, *p* < 0.01, Model 3) and LSD use (*b* = −0.632, *p* < 0.01, Model 7), meaning that the association between private insurance and lower distress is stronger among those who use these substances. Similarly to among men, public health insurance is associated with higher distress among women, but the interaction terms indicate a broader range of psychedelics amplifying this association. Specifically, the positive association between public health insurance and distress is stronger for those reporting lifetime use of classic psychedelics (*b* = 1.074, *p* < 0.001, Model 8), MDMA (*b* = 0.516, *p* < 0.05, Model 9), psilocybin (*b* = 0.852, *p* < 0.01, Model 10), peyote/mescaline (*b* = 0.921, *p* < 0.05, Model 13), and LSD (*b* = 0.902, *p* < 0.001, Model 14).

## 4. Discussion

Drawing directly from the study′s multivariate findings, this analysis examined the relationship between psychedelic use, psychological distress, healthcare access, and gender, highlighting how structural healthcare inequalities shape psychedelic-related mental health outcomes. Prior research has shown that the effects of psychedelics on mental health vary across sociodemographic factors, but this study extends that work by evaluating whether healthcare access influences these associations. Overall, the results indicate that psychedelic use affects the association between health insurance status and distress, with notable gender differences. Psychedelics reinforce the association between private insurance and lower distress, while amplifying the association between public insurance and higher distress. These effects are most pronounced among women, who show both a moderation effect for private insurance and a broader range of psychedelics intensifying distress among public insurance recipients. Among men, no significant interaction effects emerge, aligning with expectations that, due to structural advantages within healthcare, their distress levels remain less affected by psychedelic use. These findings align with the Minorities′ Diminished Psychedelic Returns (MDPR) theory, which suggests that structural inequalities—such as poverty, racism, and sexism—can limit the ability of marginalized populations to experience the same mental health benefits from psychedelics as more advantaged groups [12]. By incorporating healthcare access into the MDPR framework, this study highlights healthcare quality as a key structural factor shaping disparities in psychedelic-related mental health outcomes. While the associations are robust, they should be interpreted with caution, due to the cross-sectional nature of the data (see Limitations Section below).

As shown in the regression models, individuals with private healthcare report better mental health outcomes associated with psychedelic use. This may reflect that private healthcare systems provide conditions that facilitate more favorable psychedelic-related effects, or that individuals within these systems differ in ways that shape their ability to integrate such experiences. One possible explanation is that psychedelic use is associated with higher levels of self-regulation, stress resilience, and cognitive flexibility [13,53,83], which may enable more effective engagement with medical providers and treatment systems. These traits may be especially impactful in private healthcare settings, where patients typically have greater access to mental health services, longer consultation times, and more personalized care options [37,40]. If individuals with private healthcare are supported by more consistent and responsive medical infrastructures, they may be better positioned to translate psychedelic experiences into longer-term psychological benefits. However, these associations should be interpreted cautiously, as unmeasured factors—such as health-seeking behavior or prior experiences with therapy—may influence both healthcare access and psychedelic outcomes.

In contrast, the results indicate that individuals in public healthcare settings report no significant mental health benefits associated with psychedelic use, which may reflect differences in access to care and structural barriers to treatment [84]. Rather than functioning as a complement to ongoing mental healthcare, psychedelics in these contexts may serve as a form of self-management or coping for those navigating under-resourced systems. Individuals facing elevated distress in public healthcare environments—characterized by longer wait times, limited access to specialists, and weaker provider relationships—may be more likely to use psychedelics in response to unmet mental health needs [37,84]. This interpretation is consistent with research on negative coping under structural stress. These findings do not imply that public healthcare users inherently have worse outcomes from psychedelic use, but that contextual differences in care environments likely shape how these substances affect psychological distress. As with the other results, caution is warranted, due to the inability to determine the directionality of this association or control for potential confounding factors, such as prior trauma or self-medication patterns.

The gender-stratified results show that all significant interaction effects were observed among women, further complicating this relationship, particularly in public healthcare settings. This finding underscores the role of structural sexism in shaping both access to care and the psychological outcomes associated with psychedelic use. Among men, psychedelic use did not moderate the relationship between insurance and distress, which is itself a meaningful finding. This likely reflects the fact that, compared to women, men—especially those with private insurance—encounter fewer structural barriers, experience more expedient care, and report higher satisfaction with healthcare services overall. This aligns with research suggesting that men, particularly those with private insurance, tend to receive more expedient and less contested care, reducing the variability that psychedelics might otherwise influence [40]. In short, the absence of significant moderation effects among men may indicate that when healthcare access is more stable and less shaped by systemic inequities, the contextual impact of psychedelics is diminished. For women, however, psychedelics may contribute to improved psychological outcomes in private healthcare settings, potentially reflecting differences in how they engage with systems that have historically dismissed their symptoms [43,44,45,46,47,48]. Prior research has linked psychedelic use to increased emotional awareness, self-advocacy, and cognitive flexibility [52,53], traits that may enhance medical interactions and help to mitigate some effects of systemic bias in well-resourced settings. These findings do not suggest that psychedelics resolve gender-based disparities, but rather that existing inequities may condition how psychedelic experiences unfold. For women in public healthcare, structural barriers such as misdiagnosis, inadequate pain management, and dismissive treatment [44,47] may intensify distress and increase reliance on self-directed coping strategies, including psychedelic use. Among men, while distress levels are higher in public healthcare, psychedelics do not appear to moderate this relationship—reinforcing the idea that in the absence of intersecting gender-based barriers, psychedelic outcomes may remain unaffected by the healthcare setting. These gendered dynamics highlight the need for future research to further examine how intersecting inequalities shape both healthcare outcomes and the role that psychedelics may play within them.

Taken together, the empirical findings carry significant implications for healthcare policy, clinical practice, and public health strategy. If psychedelics are to be integrated into mental healthcare systems, they must not be treated as universal solutions, but as interventions whose outcomes are shaped by structural access to care [85,86]. The disparities observed between private and public insurance users suggest that policymakers should prioritize not only expanding access, but also ensuring equity in outcomes within public systems. One actionable strategy is to incorporate structural competency training into provider education, which equips clinicians with the ability to recognize how systemic inequalities—such as those related to insurance status, gender, and stigma—shape patient care and health outcomes [36,38]. Public health systems should also fund community-based psychedelic integration services, especially in under-resourced communities, where clinical models alone are unlikely to close access gaps [87,88,89]. The amplified distress among women in public care further underscores the need for gender equity audits in mental health delivery, including reforms in diagnostic protocols, provider accountability, and pain-management practices [44,47]. Public health campaigns and clinical rollouts of psychedelic therapies must avoid a one-size-fits-all approach, and instead prioritize equity [90]. Group therapy protocols, which have been shown to cut clinician costs by over 50% for MDMA and 34% for psilocybin, offer a scalable model to expand access for Medicaid, Medicare, and TRICARE users [91]. Addressing stigma and logistical barriers among military populations is also key to ensuring TRICARE responsiveness [92]. Additionally, MDMA-assisted therapy has demonstrated long-term cost savings for public insurers [90,93]. Without structural reform, psychedelic therapies may be disproportionately accessible to already-advantaged populations, reinforcing rather than reducing health disparities.

## 5. Limitations

This study provides valuable insights into the relationship between psychedelics, healthcare, mental health, and gender; however, several limitations must be acknowledged. First, as this study is based on cross-sectional data, it cannot establish causal relationships. The observed associations may be influenced by unmeasured factors, and the directionality remains unclear. Future research should incorporate longitudinal data to assess whether psychedelic use influences healthcare engagement, or whether healthcare access shapes psychedelic-related outcomes. Additionally, while the analysis controlled for key sociodemographic factors, other potential confounders—such as personality traits, peak psychedelic experiences, dosage, and mental health history—were not accounted for.

Another limitation is the reliance on self-reported data, which may introduce recall bias, social desirability bias, and reporting inconsistencies. Participants′ perceptions of their healthcare experiences and mental health may not align with objective measures, particularly if stigma affects disclosure. Individuals in public healthcare settings may under-report distress or dissatisfaction due to concerns about medical stigma or limited mental health awareness. Future studies should integrate objective mental health assessments and healthcare utilization data to improve reliability.

Additionally, this study does not account for access to psychedelic integration services, which may influence the sustainability of psychedelic-related benefits. Individuals with private healthcare may have greater access to therapy, peer support, and harm-reduction services, while those in public healthcare may lack structured support for processing psychedelic experiences. Without integration services, even beneficial experiences may not lead to long-term improvements, potentially reinforcing disparities. Future research should explore how access to these resources affects psychedelic-related health outcomes.

This study also lacks data on set and setting, which are critical to psychedelic experiences. The psychological state of the user, social support, and their environment all influence outcomes, yet this study does not distinguish between structured therapeutic use and unregulated, distress-driven use. Similarly, differences in psychedelic substances and frequency of use are not considered. Psychedelics vary widely in their effects, duration, and intensity, making it unclear whether the observed associations differ based on specific substances. Future research should examine how both the context and substance type shape the relationship between psychedelics and mental health across healthcare settings.

## 6. Conclusions

This study highlights the role of healthcare disparities in shaping psychedelic-related health outcomes. While private healthcare is associated with better outcomes, individuals relying on public healthcare—who often experience the highest levels of distress—do not appear to receive the same benefits. In fact, for women in public healthcare, the structural barriers they face may further intensify distress, suggesting that those who are most vulnerable and in greatest need of relief may see little-to-no benefit from psychedelic use. These findings indicate that while psychedelics may complement existing healthcare systems, they are unlikely to compensate for deeply entrenched inequities. Rather than assuming that psychedelics can serve as a universal solution, policy efforts should focus on expanding equitable access to quality care, improving gender-inclusive medical training, and addressing disparities between public and private systems. Future research should further examine how sociocultural conditions shape psychedelic experiences beyond clinical models that assume universal benefits. As psychedelics become more widely available, understanding the structural barriers that influence their health effects will be essential for ensuring equitable access and outcomes.

## Figures and Tables

**Figure 1 healthcare-13-01158-f001:**
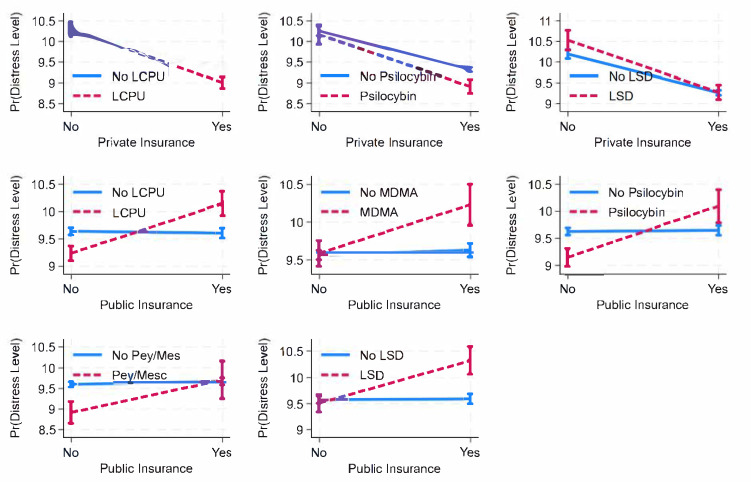
Predicted margins of private or public health insurance (x) psychedelic use on psychological distress among total population, with 95% CIs. Source: National Survey of Health and Social Behaviors, 2008-2019. Note: Based on Models 1, 3, 7, 8, 9, 10, 13, and 14, Appendix A, and multinomial ordinary least square regression model predicting distress level in last 30 days.

**Figure 2 healthcare-13-01158-f002:**
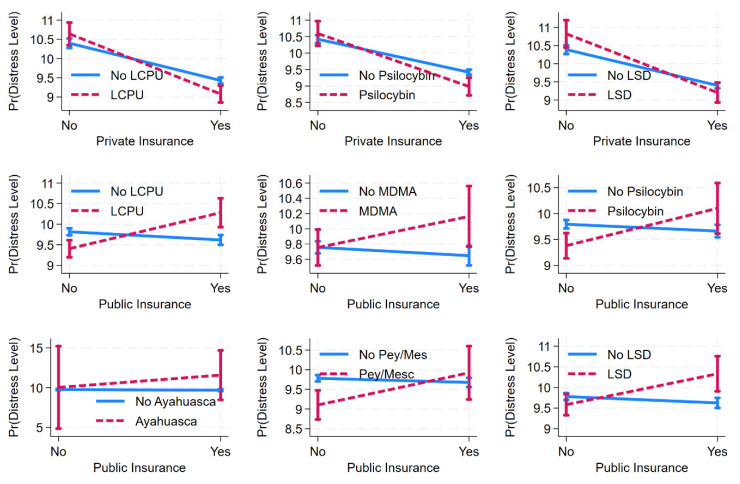
Predicted margins of private or public health insurance (x) psychedelic use on psychological distress among women, with 95% CIs. Source: National Survey of Health and Social Behaviors, 2008-2019. Note: Based on Models 1, 3, 7, 8, 9, 10, 12, 13, and 14, Appendix A, and multinomial ordinary least square regression model predicting distress level in last 30 days.

**Table 1 healthcare-13-01158-t001:** Descriptive statistics for dependent variables, independent variables, and control variables (2008–2019) (weighted).

	Mean	SD	n	%/min–max
Key Predictor Variables				
Psychological distress	9.59	6.08	161,537	0–24
Women			250,942	51.77
No health insurance			63,390	13.08
Private insurance			321,954	66.42
Public insurance			157,741	32.54
Other health insurance			9635	1.99
Lifetime psychedelic use				
MDMA			34,224	7.06
Psilocybin			38,475	7.94
DMT			461	0.10
Ayahuasca			43	0.01
Peyote/mescaline			13,091	2.70
LSD			35,666	7.36
Classic psychedelic use			66,854	13.79
Control Variables				
Age	8.67	2.31	484,732	1–24
Education	2.75	1.03	484,732	1–4
Family income	4.96	2.02	484,732	1–7
Marital status				
Single			134,007	27.65
Married			254,501	52.50
Widowed			28,908	5.96
Divorced/separated			67,316	13.89
Children	0.54	0.92	484,043	0–3
Race				
White			318,261	65.66
Black			56,784	11.71
Hispanic			73,291	15.12
Asian			24,930	5.14
Native American			2543	0.52
Hawaiian			1735	0.36
Multiracial			7189	1.48
Religious salience	4.92	2.60	472,653	0–9
Religious attendance in days	1.89	0.92	480,882	0–5
Lifetime drug use				
Cocaine			78,221	16.14
Stimulants			49,164	10.14
Sedatives			40,022	8.26
Tranquilizer			79,692	16.44
Inhalants			42,851	8.85
Pain-relievers			174,374	35.97
Heroine			9528	1.97
Marijuana			225,161	46.45
PCP			12,643	2.61
MDMA/ecstasy			44,156	9.12
Tobacco			277,750	57.30
Age of first alcohol use	2.88	1.16	484,732	1–5
Thrill-seeking behavior	1.62	0.72	481,653	1–4

Source: 2008–2019 National Survey of Drug Use and Health, *n =* 484,732.

## Data Availability

The National Survey of Drug Use and Health (NSDUH) is public-use data, and is available on their homepage: https://www.samhsa.gov/data/data-we-collect/nsduh-national-survey-drug-use-and-health/datafiles/2002-2019, 11 January 2024.

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
