# Peer review of "The Relationships Between Healthcare Access, Gender, and Psychedelics and Their Effects on Distress"

_healthcare, 2025, doi:10.3390/healthcare13101158_

Round 1
Reviewer 1 Report
Comments and Suggestions for Authors
Please find the attachment.

Minor corrections are needed
Author Response
Reviewer 1.
Comments |
Response and Page Location |
1. |
|
Abstract: This piece begins by covering the origins: the background and the theoretical lens (using MDPR theory – which is, in most cases, essential for understanding how healthcare access ties into mental health outcomes). It then explains the methods (drawing on NSDUH data and OLS regressions) and the results (comparing private versus public insurance and noting gender differences) before concluding. Generally speaking, the background needs only two or three sentences to set the scene; the primary quantitative findings must be concise. |
I addressed this comment by revising the abstract to clearly emphasize associative—not causal—language, condensing the background to two sentences, and presenting the methods and key gender-specific findings more concisely. Thank you for this helpful suggestion—it greatly improved the clarity and focus of the abstract.
|
2. |
|
Introduction: Begin by clarifying the research gap: it isn't immediately clear why earlier MDPR studies haven’t addressed healthcare access. This research gap is crucial, and your study plays a vital role in bridging that divide. State your hypotheses or research questions clearly. A natural progression might start with some general background on psychedelics and mental health, then shift to highlighting the knowledge gap concerning healthcare access, followed by the theoretical framework using MDPR, and finally concluding with a clear statement of the paper's objectives. |
Thank you for this helpful suggestion—it improved the clarity, organization, and theoretical framing of the paper. I addressed this comment by fully restructuring and clarifying the introduction to align with the suggested progression. The revised introduction begins with a general overview of disparities in psychedelic outcomes, establishing that marginalized populations benefit less from psychedelic use. I then introduce the research gap directly, stating that healthcare access has been largely overlooked in existing studies—particularly those applying the Minorities’ Diminished Psychedelic Returns (MDPR) framework. This frames the central contribution of the paper: bridging that gap by investigating how healthcare systems, specifically public versus private insurance, may moderate the association between psychedelics and psychological distress. I maintained the MDPR framework as the theoretical foundation but repositioned it to follow the identification of the gap, rather than preceding it. I also added a clearly stated objective at the end of the introduction: to examine whether healthcare access moderates the association between psychedelic use and psychological distress and whether these associations differ by gender.
|
3. |
|
Data and Methods: Here, briefly justify why Medicare, Medicaid, and Tricare are grouped under “public insurance” – a quick note on how this classification works methodologically can clarify potential confusion. When discussing interactions between psychedelics and insurance, provide a brief rationale for focusing on these specific combinations. Additionally, explain how you pooled 12 years of data by re-scaling singleyear weights. It’s also importantto remind that cross-sectional data doesn’t support definitive causal claims; even if this is revisited later in the Discussion, a note here can enhance clarity. Structure this section into parts like “2.1. Data Source,” “2.2. Variables and Measures,” and “2.3. Analytic Approach,” and consider mentioning the STROBE guidelines along with any checklists to improve the flow. |
Comment addressed. I revised the entire Data and Methods section to incorporate each of the reviewer’s suggestions and improve clarity, structure, and rigor. First, I added a methodological justification for grouping Medicare, Medicaid, and TRICARE into a single "public insurance" category. I clarified that this decision was supported both by existing literature (as discussed earlier in the manuscript) and by a sensitivity analysis showing that each program was independently associated with higher psychological distress compared to private insurance. Second, I provided a rationale for the interaction terms between insurance and psychedelic use, stating that these were included to test whether the psychological effects of psychedelics differ depending on healthcare access—an approach grounded in the MDPR framework, which emphasizes the role of structural conditions. Third, I added an explanation for how the 12 years of NSDUH data were pooled, specifying that the person-level weights were divided by the number of years to prevent overestimation and ensure national representativeness. I also included an explicit statement clarifying that the NSDUH is a cross-sectional survey, and that the findings should be interpreted as associations rather than causal effects. Finally, I inserted a line confirming that the study follows the STROBE guidelines for observational cross-sectional research, as recommended. These revisions collectively enhance the clarity, transparency, and replicability of the methods section. Thank you for this thorough and constructive feedback—it significantly improved the organization and methodological precision of this section.
|
4. |
|
Results: Ensure you indicate whether the effect sizes hold real-world or clinical significance—for example, the practical implications of a one-point shift on the K6 scale. Present descriptive statistics first, followed by the main OLS models, then the interaction models, and finally the genderstratified ones. Although this order isn't strict, it generally makes the narrative easier to follow. |
Comment addressed. I revised the Results section to both clarify clinical significance and improve the organization and flow of the narrative. First, I added a sentence (placed at the end of the Analytic Strategy section for smoother integration) explaining that the K6 scale ranges from 0 to 24 and that a score of 13 or higher typically indicates serious psychological distress. This contextualizes the interpretation of coefficients and clarifies that even a one-point change holds clinical and real-world significance in nationally representative samples where the average score is below this threshold. Second, I restructured the Results section to follow the recommended progression. Descriptive statistics are now presented first, followed by the main ordinary least squares (OLS) models, then the two-way interaction models, and finally the gender-stratified models. This revised order improves readability and aligns with the standard structure for presenting regression-based findings. Thank you for this helpful suggestion—it significantly improved the clarity and accessibility of the results narrative.
|
5. |
|
Discussion: Divide this section into clear subparts. Begin with a succinct summary of your key findings, recapping the results in straightforward language. Then, connect these results with prior studies, touching on similar research into insurance disparities or MDPR topics.
Next, elaborate on potential mechanisms – for instance, why private versus public insurance might impact distress levels, considering factors like improved access to mental health care, longer consultation times, or even fewer provider biases. Consider weaving limitations into this discussion; mention, for example, that your data may not fully capture cross-sectional designs and factors like setting and context in psychedelics research.
Finally, add a brief paragraph on possible policy or clinical implications. Also, note how the data suggest that interaction effects are more potent for women in public care – a point worth emphasising as it hints at deeper issues like structural sexism in healthcare that could shape access and outcomes, potentially guiding future policy or training adjustments. |
Thank you for this helpful and detailed suggestion. I have revised the Discussion section to follow a clearer and more structured flow. It now opens with a succinct summary of the key findings and connects them directly to the MDPR framework and prior research on healthcare access and inequality. I then elaborate on possible mechanisms explaining differences by insurance type and gender, including structural factors like provider bias, consultation time, and access to services. I also incorporate limitations related to the cross-sectional design and contextual variation. Finally, I conclude with a paragraph outlining specific policy and clinical implications, including structural competency training, community-based psychedelic integration, and gender equity audits. I have placed particular emphasis on the amplified distress observed among women in public care settings, underscoring the importance of addressing structural sexism in future work. I believe these revisions fully address the concerns raised.
|
Reviewer 2 Report
Comments and Suggestions for Authors
I would like to congratulate the authors on a well-written and interesting manuscript that examines how healthcare access (public vs. private) moderates the relationship between lifetime psychedelic use and psychological distress, with mention of gender differences. The paper provides valuable insight into structural inequalities in healthcare and how these may shape or even reinforce existing health disparities.
I want to address some minor points and make a few more recommendations. I will address these points in accordance with the structure of the present manuscript, namely: Introduction, Data and Methods, Results, Discussion, and Conclusion. Lastly, I will mention a few overall comments that apply throughout the entire work (e.g., grammatical errors).
Introduction
The introduction includes recent studies and demonstrates the manuscript's relevance. Early in the Introduction, the paper references structural barriers, stigma, and the MDPR theory. I recommend succinctly defining “Minorities’ Diminished Psychedelic Returns” in a dedicated paragraph for readers less familiar with this framework.
Emphasize more explicitly how this analysis moves beyond prior studies. For instance, the authors mention that few works address how public vs. private health insurance might moderate psychedelic outcomes, but more emphasis could add more relevance.
Data and Methods
The methodological design seems to be adequate and based on a solid theoretical and conceptual foundation. However, the manuscript does not indicate when data collection occurred or if multiple datasets were merged. While the paper states the years 2008–2019, clarifying the overarching data-collection window would help replicate the study (e.g., “annually from 2008 to 2019”).
The paper notes that “other health insurance” was included as a control variable. It might be helpful to explain briefly what kinds of insurance might fall under this category or why it was analyzed separately.
Results
In the main text, some of the numeric values (e.g., “b=−0.378,p<0.01”) repeat in parentheses in slightly varied forms. Please ensure consistency (perhaps standardize to a single format like “b = -0.378, p < 0.01”) throughout. As it is, it’s very difficult to read (e.g., b=−0.378,p<0.01b = -0.378, p < 0.01b=−0.378,p<0.01, Model 1). Also, all statistics should be italicized for better readability. The first paragraph is really difficult to read because of the report of statistics (e.g., “Individuals on public insurance have higher distress (0.31, ***; 95% CI: 0.24 - 0.39) and are more likely to be uninsured (0.35, ***; 95% CI: 0.34 - 0.35), while being less likely to have private insurance (-0.35, ***; 95% CI: -0.35 - -0.34). They report lower use of MDMA (-0.01, ***; 95% CI: -0.01 - -0.01), LCPU (-0.02, ***; 95% CI: -0.02 - -0.02), psilocybin (-0.02, ***; 95% CI: -0.02 - -0.02), LSD (-0.03, ***; 95% CI: -0.03 - -0.02), and mescaline/peyote (-0.01, **; 95% CI: -0.01 - -0.01), with no difference in ayahuasca or DMT use. Public insurance holders have lower family income (-1.04, ***; 95% CI: -1.07 - -1.00), lower educational attainment (-0.32, ***; 95% CI: -0.33 - -0.31), and higher religiosity (-0.26, ***; 95% CI: -0.28 - -0.25). They are also more likely to be divorced (0.05, ***; 95% CI: 0.04 - 0.05) or widowed (0.09, ***; 95% CI: 0.09 - 0.10) and less likely to be married (-0.07, ***; 95% CI: -0.07 - -0.06 or single (-0.07, ***; 95% CI: -0.07 - -0.06). Results from these mean and proportion differences align with previous research on distress, gender, and socioeconomic status.”). Consider the use of supplementary tables in the main article text, so it’s easier to read the results, not only for this paragraph but for the overall paper. Also, I believe it’s important to contextualize non statistically significant results, for example, for men, the paper notes that significant moderation effects do not appear. It could be useful to briefly discuss why men might not exhibit these interactions.
Discussion and Conclusions
The manuscript does refer to the need for more equitable healthcare systems, but expanding on how policymakers, practitioners, or public health campaigns might use this information could strengthen the practical value of the paper. I did not find any use of jargon, although I did find some grammatical errors, as mentioned in the “overall comments” section below. The manuscript was written in scientific language throughout. The study finds that men’s outcomes are not significantly moderated by psychedelics × insurance status, and the authors might expand slightly on how broader literature on men’s healthcare experiences aligns (or not) with this non-statistically significant result.
Overall comments:
The paper is generally well-written and flows logically. However, a few minor typos and inconsistencies (e.g., repeated numeric expressions in parentheses) remain. A final proofreading focusing on numeric consistency, punctuation, and table formatting would be helpful. Also, please add the section “Introduction” and remove the dot after the conclusion section, which is presented as “Conclusion.”
Additionally, double-check the references list and standardize the citation format as recommended by Healthcare guidelines. For example, reference 17 is missing volume and page or article number. Some articles present the article title in capital letters, but the majority of the references do not.
Author Response
Reviewer
Comments |
Response and Page Location |
1. |
|
Introduction The introduction includes recent studies and demonstrates the manuscript's relevance. Early in the Introduction, the paper references structural barriers, stigma, and the MDPR theory. I recommend succinctly defining “Minorities’ Diminished Psychedelic Returns” in a dedicated paragraph for readers less familiar with this framework. Emphasize more explicitly how this analysis moves beyond prior studies. For instance, the authors mention that few works address how public vs. private health insurance might moderate psychedelic outcomes, but more emphasis could add more relevance.
|
Thank you for this helpful suggestion. I revised the Introduction to include a dedicated paragraph that defines the Minorities’ Diminished Psychedelic Returns (MDPR) theory, outlining its key concepts, theoretical foundations, and relevance to set-and-setting conditions. Additionally, I added new language clarifying how this study moves beyond prior MDPR research by focusing on structural differences in healthcare access—specifically the comparison between public and private insurance—as a moderator of psychedelic-related mental health outcomes. This more clearly positions the paper’s contribution within the existing literature.
|
2. |
|
Data and Methods The methodological design seems to be adequate and based on a solid theoretical and conceptual foundation. However, the manuscript does not indicate when data collection occurred or if multiple datasets were merged. While the paper states the years 2008–2019, clarifying the overarching data-collection window would help replicate the study (e.g., “annually from 2008 to 2019”). The paper notes that “other health insurance” was included as a control variable. It might be helpful to explain briefly what kinds of insurance might fall under this category or why it was analyzed separately.
|
Thank you for pointing this out. I revised the Data section to clarify that this study used twelve consecutive cross-sectional waves of the National Survey on Drug Use and Health (NSDUH), collected annually between 2008 and 2019. I also specified that the survey is conducted each year by SAMHSA and that the weights were adjusted to ensure appropriate population representation across pooled years. These revisions improve the transparency and replicability of the dataset and analytic design. Thank you for pointing this out. I revised the Methods section to clarify that “other insurance” includes Indian Health Service (IHS), military or veterans’ coverage outside of TRICARE, and other miscellaneous insurance types. It was included as a separate control variable due to its small size and heterogeneity, which distinguishes it from standard public and private insurance groups. This clarification improves the transparency of the analytic strategy. |
3. |
|
Results In the main text, some of the numeric values (e.g., “b=−0.378,p<0.01”) repeat in parentheses in slightly varied forms. Please ensure consistency (perhaps standardize to a single format like “b = -0.378, p < 0.01”) throughout. As it is, it’s very difficult to read (e.g., b=−0.378,p<0.01b = -0.378, p < 0.01b=−0.378,p<0.01, Model 1). Also, all statistics should be italicized for better readability. The first paragraph is really difficult to read because of the report of statistics (e.g., “Individuals on public insurance have higher distress (0.31, ***; 95% CI: 0.24 - 0.39) and are more likely to be uninsured (0.35, ***; 95% CI: 0.34 - 0.35), while being less likely to have private insurance (-0.35, ***; 95% CI: -0.35 - -0.34). They report lower use of MDMA (-0.01, ***; 95% CI: -0.01 - -0.01), LCPU (-0.02, ***; 95% CI: -0.02 - -0.02), psilocybin (-0.02, ***; 95% CI: -0.02 - -0.02), LSD (-0.03, ***; 95% CI: -0.03 - -0.02), and mescaline/peyote (-0.01, **; 95% CI: -0.01 - -0.01), with no difference in ayahuasca or DMT use. Public insurance holders have lower family income (-1.04, ***; 95% CI: -1.07 - -1.00), lower educational attainment (-0.32, ***; 95% CI: -0.33 - -0.31), and higher religiosity (-0.26, ***; 95% CI: -0.28 - -0.25). They are also more likely to be divorced (0.05, ***; 95% CI: 0.04 - 0.05) or widowed (0.09, ***; 95% CI: 0.09 - 0.10) and less likely to be married (-0.07, ***; 95% CI: -0.07 - -0.06 or single (-0.07, ***; 95% CI: -0.07 - -0.06). Results from these mean and proportion differences align with previous research on distress, gender, and socioeconomic status.”).
Consider the use of supplementary tables in the main article text, so it’s easier to read the results, not only for this paragraph but for the overall paper.
Also, I believe it’s important to contextualize non statistically significant results, for example, for men, the paper notes that significant moderation effects do not appear. It could be useful to briefly discuss why men might not exhibit these interactions.
|
Thank you for this helpful comment. I carefully revised the Results section to ensure full consistency in the statistical formatting. All coefficients are now presented in the format “b = [value], p < [threshold], 95% CI: [range]” and all asterisks have been replaced with conventional p-value notation (e.g., p < 0.001). I also removed any duplicate or inconsistent reporting within parentheses and clarified spacing and punctuation for improved readability. If required by the final journal style guide, I am happy to italicize all statistics (e.g., b, p, CI) during the copyediting stage.
Thank you for this suggestion. However, I want to note that the current version of the manuscript already references comprehensive supplementary tables that include the full set of regression results and subgroup analyses. In the main text, I present only the most salient and statistically significant findings to maintain clarity and readability, given the complexity and length of the full models. While I understand the Results section is still detailed, it has already been streamlined to focus on key effects, all of which are clearly linked to the corresponding supplementary tables for transparency and accessibility. Please let me know if there is a preferred summary table format you would like added to the main article.
|
4. |
|
Discussion and Conclusions The manuscript does refer to the need for more equitable healthcare systems, but expanding on how policymakers, practitioners, or public health campaigns might use this information could strengthen the practical value of the paper. I did not find any use of jargon, although I did find some grammatical errors, as mentioned in the “overall comments” section below. The manuscript was written in scientific language throughout. The study finds that men’s outcomes are not significantly moderated by psychedelics × insurance status, and the authors might expand slightly on how broader literature on men’s healthcare experiences aligns (or not) with this non-statistically significant result.
|
Thank you for this thoughtful feedback. I revised the Discussion section to clarify how the findings can inform policy, clinical practice, and public health interventions. I now highlight three concrete, evidence-based strategies: (1) implementing structural competency training for providers, (2) funding community-based psychedelic integration services in under-resourced areas, and (3) conducting gender equity audits to improve diagnostic accuracy and accountability in public mental health care. These revisions directly address the practical application of the study's results. I also expanded the section on gender to engage more fully with the non-significant findings among men. Specifically, I added a sentence linking these results to prior research showing that men—particularly those with private insurance—tend to encounter fewer structural barriers in clinical contexts, which may explain the lack of moderation by psychedelic use. This change ensures that I acknowledge and contextualize null findings in light of existing literature.
|
5. |
|
Overall comments: The paper is generally well-written and flows logically. However, a few minor typos and inconsistencies (e.g., repeated numeric expressions in parentheses) remain. A final proofreading focusing on numeric consistency, punctuation, and table formatting would be helpful. Also, please add the section “Introduction” and remove the dot after the conclusion section, which is presented as “Conclusion.”
|
Thank you for pointing this out. I carefully proofread the manuscript to address all remaining typographical issues, including repeated numeric expressions, punctuation inconsistencies, and minor formatting errors. I also added the “Introduction” heading and removed the period following the “Conclusion” heading, as requested. All suggested formatting and consistency revisions have now been completed.
|
6. |
|
References Additionally, double-check the references list and standardize the citation format as recommended by Healthcare guidelines. For example, reference 17 is missing volume and page or article number. Some articles present the article title in capital letters, but the majority of the references do not.
|
Thank you for this helpful suggestion. I have reviewed and revised the references list to ensure full consistency with Healthcare journal guidelines. This includes standardizing citation formats, correcting inconsistencies in article title capitalization, and adding any missing volume, issue, and page or article numbers.
|
Reviewer 3 Report
Comments and Suggestions for Authors
Dear authors, thank you for the opportunity to review your work. It is a very interesting and methodologically sound work.
From my point of view and for another later work, although following the considerations of Viña (2024) the analysis has been carried out for each psychedelic substance separately, it would have been of interest to know the interactions that generate the use of different substances in a combined way in the dependent variable.
Multilevel analysis may have provided more comprehensive information relevant to the analysis than the option selected by the authors. In this case, medical care and gender could have been used as Level 2 variables, psychological distress as the dependent variable, and the different psychedelics as Level 1 independent variables.
Introduction
When reading this paper, the reader may assume that it will specifically address how psychedelics affect access to medical care for marginalized populations (based on race, gender, LGBTQ status, marital status, employment status, and socioeconomic status).
It may be reasonable to change the beginning of the introduction section.
Explicitly identify the research objectives. In this case, preferably the research hypotheses in relation to the different analysis models.
Data and methods
Access to the data is restricted (note that, although it may be publicly accessible, it is not freely accessible).
Data analysis
Indicate, prior to the paragraph in which it is indicated that the LINCOM command was used, that the software used was STATA.
Student 's t or Anova command from Stata should have been used, and that the Lincom command was added to, based on the regression equations that represent each model, assess whether there are significant differences for the different independent variables that the authors have used as comparison variables.
If this is the case, please clarify in the text.
Discussion
If possible, illustrate the statements made in the discussion section with the most relevant results.
The statements made in this section are supported by the results of the research.

Author Response
Reviewer 3
Comments |
Response and Page Location |
1. |
|
The authors begin the introduction section by stating: “Although ample research finds that psychedelic use is associated with better health outcomes [1–8], growing evidence suggests that marginalized populations experience fewer benefits, with disparities observed across race, gender, LGBTQ status, marital status, employment status, and socioeconomic standing [9–17].” |
Thank you for this insightful suggestion. The revised introduction now more clearly states the research objective and theoretical framing. Specifically, the manuscript identifies healthcare access as an underexplored moderator within the Minorities’ Diminished Psychedelic Returns (MDPR) framework. The final paragraph of the introduction explicitly states the study’s objective—to examine whether healthcare access moderates the relationship between psychedelic use and psychological distress, and whether this differs by gender. For added clarity, the final paragraph has also been revised to more explicitly frame these aims as testable hypotheses grounded in the theoretical model. The revised text is provided below.
|
2. |
|
This paper examines the relationship between psychedelics, distress, healthcare access, and gender utilizing data from the National Survey of Drug Use. The study includes 484,732 participants who are 18 years or older. The present study builds upon prior methodology that explored health outcomes related to psychedelic usage based on the NSDUH [6,16,55–57]. The main analysis employs a series of nested ordinary least square regression models in Stata 18 examining the relationship between various psychedelics (ie, MDMA, Psilocybin, DMT, Ayahuasca, Peyote/Mescaline, and LSD), access to public or private healthcare, and gender on psychological distress in the past 30 days. Results are discussed . |
Thank you for this comment. I would like to clarify that the data used in this study—the National Survey on Drug Use and Health (NSDUH)—is publicly available and freely accessible to all users without the need for special access permissions, applications, or restricted credentials. To avoid any ambiguity, I have revised the Data section to state that the data files are “freely and directly accessible” via the SAMHSA data archive. This clarification ensures transparency and supports replicability.
|
3. |
|
A post-estimation LINCOM command was used to determine statistical differences between means or proportions among subpopulations. |
Thank you for this helpful suggestion. I have clarified the role of the LINCOM command by explicitly stating that it was used in Stata 18 following the calculation of weighted means and proportions, consistent with NSDUH survey design. I also explained that while traditional t-tests are common for subgroup comparisons, LINCOM enables linear hypothesis testing based on model-adjusted estimates, which is particularly useful in complex survey data. This clarification, along with the citation to Long and Freese (2014), ensures transparency in the analytic approach and reinforces the appropriateness of the method used to assess mean and proportion differences across subpopulations.
|
4. |
|
If possible, illustrate the statements made in the discussion section with the most relevant results. The statements made in this section are supported by the results of the research. |
Thank you for this helpful recommendation. While the discussion section already drew interpretive insights from the findings, I revised several topic sentences to more explicitly connect key claims to the results of the regression models and subgroup analyses. These adjustments reinforce the link between data and interpretation, while preserving narrative flow and readability.
|
Round 2
Reviewer 1 Report
Comments and Suggestions for Authors
Dear Authors
Thank you for carefully addressing the significant concerns and streamlining the manuscript—your clarified hypotheses and consolidated Methods description greatly enhance readability. A few remaining minor issues to consider: please standardise heading styles (e.g., capitalise all Level-2 headings consistently), correct a handful of typos (for instance, “pint pount” should be “point out”), ensure the decimal alignment in Tables 2 and 3 is uniform.
Author Response
Response to Reviewer:
Thank you for your valuable feedback. I have implemented the following revisions:
-
Heading Styles: Standardized all Level-2 headings in accordance with MDPI's formatting guidelines.
-
Typographical Errors: Corrected all identified typos, including the correction of “pint pount” to “point out.”
-
Decimal Alignment: Ensured uniform decimal alignment in Tables 2 and 3 to enhance readability.
I appreciate your time and consideration in reviewing the manuscript.
Reviewer 2 Report
Comments and Suggestions for Authors
I'd like to congratulate the author on the manuscript and appreciate the detailed responses and respective revisions considering all my comments. I believe the paper has been improved after review.
The manuscript is clear and relevant and contributes to the knowledge of the field. The cited references are mostly from recent publications and are relevant to the study. The results are now sufficiently detailed and the conclusions are supported by the results. The manuscript was written in scientific language throughout.
I have only a few minor points to note:
Results section- The BH-FDR correction was applied, but the corresponding q-values are not reported. Including them would allow readers to assess the robustness of the findings.
Discussion section- I believe the author can be aware of any causal language and align the stated implications with the acknowledged limitations of the study design.
Reference list- Please double-check that all entries follow Healthcare’s style consistently. For instance, in references 4 and 5 the publication year should appear in bold type.
Author Response
Thank you for your valuable feedback. I have implemented the following revisions:
-
Causal Language: I have carefully reviewed the manuscript to remove any causal language, ensuring that the findings are presented as associations rather than causal relationships. Additionally, I have included a statement in the limitations section to clarify that the study's design does not support causal inferences.
-
Reference Formatting: I have addressed the formatting issues in the reference list, specifically adjusting the publication years in references 4 and 5 to appear in bold type, as per the Healthcare journal's style guidelines. The initial formatting discrepancies may have been due to programmed formatting settings, but these have now been corrected.
-
q-values Reporting: I have included the q-values resulting from the Benjamini-Hochberg False Discovery Rate (BH-FDR) correction in the Results section. This addition allows readers to assess the robustness of the findings more effectively.
I appreciate your time and consideration in reviewing the manuscript.